# Serum Amyloid A Concentrations of Healthy and Clinically Diseased Japanese Black Breeding Cattle—Preliminary Measurements for Determining the Cut-Off Concentrations

**DOI:** 10.3390/vetsci9050198

**Published:** 2022-04-21

**Authors:** Urara Shinya, Yuka Iwamura, Osamu Yamato, Dhidhi Pambudi, Oky Setyo Widodo, Masayasu Taniguchi, Mitsuhiro Takagi

**Affiliations:** 1Kagoshima Agriculture Mutual Aid Association Soo Branch, Soo 899-8212, Japan; urara@nosai-soo.com (U.S.); yuka@nosai-soo.com (Y.I.); 2Joint Faculty of Veterinary Medicine, Kagoshima University, Kagoshima 890-0065, Japan; osam@vet.kagoshima-u.ac.jp; 3Department of Mathematics Education, Faculty of Teacher Training and Education, Sebelas Maret University, Surakata 57126, Indonesia; dhidhipambudi@staff.uns.ac.id; 4Joint Graduate School of Veterinary Sciences, Yamaguchi University, Yamaguchi 753-8515, Japan; oky.widodo@fkh.unair.ac.id; 5Department of Animal Husbandry, Faculty of Veterinary Medicine, Airlangga University, Surabaya 60115, Indonesia; 6Laboratory of Theriogenology, Joint Faculty of Veterinary Medicine, Yamaguchi University, Yamaguchi 753-8515, Japan; masa0810@yamaguchi-u.ac.jp

**Keywords:** cut-off concentrations, Japanese Black, serum amyloid A

## Abstract

The present study aimed to compare serum amyloid A (SAA) concentrations of Japanese Black (JB) breeding cows in both clinically normal and diseased cows diagnosed by veterinarians using modified latex agglutination turbidimetric immunoassay (LATIA) to determine the cut-off values for healthy and diseased JB cows. For the comparison, a total of 289 serum samples of healthy cows without any clinical symptoms intended for the metabolic profile test and 66 serums from diseased cows clinically diagnosed by veterinarians were measured for the SAA concentrations. A significant difference (*p*-value = 6.68 × 10^−29^) was observed in the mean SAA concentrations between the healthy (2.8 ± 3.2 mg/L) and diseased (54.8 ± 76.8 mg/L) groups, and the median concentrations of the healthy and diseased groups were 1.5 mg/L and 31.2 mg/L, respectively. Finally, the cut-off SAA concentrations at each probability were 2.9 mg/L (*p* = 0.05), 5.7 mg/L (*p* = 0.1), 13.7 mg/L (*p* = 0.5), and 21.8 mg/L (*p* = 0.9), respectively, and 6.5 mg/L (*p* = 0.122) based on evaluation performed using the receiver operating characteristic curve. The results indicated that, with the practical application of the obtained cut-off value, the measurement of SAA concentrations for JB breeding cows with LATIA could be potentially beneficial in the early evaluation of inflammatory diseases in JB breeding cows and possibly useful in the prevention of not only metabolic diseases but also non-nutritional diseases during the perinatal period of JB breeding cows.

## 1. Introduction

For optimal reproductive management of breeding beef cattle, the following two strategies are known to increase productivity: minimizing the calving-to-conception interval and producing one calf per year per cow. Moreover, several factors, such as calf sucking duration, genetic variation, and maternal nutrient intake level, affect the postpartum reproductive performance of breeding beef cows [1,2,3]. Thus, the general health condition of each cow, including normal postpartum recovery of the genital tracts, is essential for achieving the ideal productivity of breeding beef cows. Therefore, additional biomarkers associated with non-physiological, non-genetic, and non-nutritional stressors may be useful in the monitoring of postpartum breeding beef cows [4]. Previous reports indicated that acute phase proteins (APPs) are not only useful for monitoring inflammatory processes for diagnostic and prognostic purposes but also for analyzing various non-inflammatory conditions, such as pregnancy, parturition, metabolic diseases, and both environmental and management stress [4,5]. Therefore, APPs can be used as clinical parameters to indicate the health condition of postpartum cows under several pathological conditions and could be beneficial in not only monitoring postpartum reproductive failure but also identifying subclinical disorders for breeding cattle herds. Serum amyloid A (SAA), similar to any other APP, such as C-reactive protein and haptoglobin, is now clinically used as an inflammatory marker in dairy and beef cattle practice [6,7,8,9,10]; moreover, SAA reportedly responded most rapidly to infection. Therefore, SAA can be used as a potential marker in distinguishing the severity of inflammation [11].

Several quantitative analysis methods, such as indirect enzyme-linked immunoassay, non-competitive enzyme immunoassay, and latex agglutination test, have been reported so far [5,12] for the measurement of SAA concentrations in veterinary practice. However, due to its time-consuming and cost reasons, the SAA analysis in cattle has been restricted to experimental purposes only. Recently, however, the diagnostic utility of SAA measurements of cattle diagnosed with mastitis was reported upon applying the reagent developed for cattle SAA measurement with a latex agglutination turbidimetric immunoassay (LATIA) using an automated clinical chemistry analyzer [12]. We also recently reported significant differences in the SAA concentrations between healthy and clinically pathological groups upon applying the LATIA methods for Holstein dairy cattle and of Japanese Black (JB) calves, thereby indicating the usefulness to estimate/distinguish the inflammatory status and prognosis of the animals with or without (subclinical phase) clinical symptoms based on the SAA concentrations [13,14]. The advantages of SAA analysis using the LATIA system are (1) possible measurement of many samples at a reasonable price in one assay, and (2) detection of the inflammatory status of not only healthy and diseased cattle but also cattle without clinical symptoms (subclinical phase). Therefore, in order to monitor potential diseases in clinical cattle practice using the established LATIA method, the possible cut-off value of the SAA concentration between healthy cattle and cattle suspected with a potential inflammatory status, including subclinical condition, could be beneficial in disease prevention and early treatment of the cattle herd, especially the herd with routinely introduced metabolic profile test (MPT). However, to the best of our knowledge, no study exists on the SAA cut-off values in healthy cattle and those with a clinically pathological condition, especially in JB breeding cows. SAA concentration may be considered an additional clinical parameter for the early detection of diseased JB breeding cattle and determination of the prognosis, which could be important in improving the productivity and fertility of the herds. Therefore, the present study aimed to obtain the cut-off values of the SAA concentration in JB breeding cattle from the SAA concentrations measured by the LATIA method derived from clinically diagnosed healthy and diseased cattle.

## 2. Materials and Methods

The experiments were conducted according to the regulations concerning the protection of experimental animals and the guidelines of Yamaguchi University, Yamaguchi, Japan (No. 40, 1995, approved on 27 March 2017).

### 2.1. JB Breeding Herds and Cows

A total of 21 private JB breeding herds under the jurisdiction of Soo Veterinary Clinical Center, Soo Agriculture Mutual Aid Association, Kagoshima Prefecture, Japan, were identified for this study. Cows included in this study (*n* = 289) were of different ages and in different stages of the reproductive cycle without any visible signs of clinical disease intended for the routine MPT in our clinical laboratory. All the cows were housed indoors and fed roughage collected from the home pastureland and supplementary concentrate purchased from various feed companies; however, feeding and management varied among the herds. Blood samples from the healthy cows were collected from the jugular vein of the cows of each herd after the morning feeding. Cows that were examined by a veterinarian and diagnosed as diseased with clinical symptoms based on the farm’s request for medical treatment from 53 private JB breeding herds were defined as the diseased cow group, and blood samples (*n* = 66) of these cows were collected to confirm the clinical diagnosis before or after their medical treatments as shown in Table 1. After collecting the blood samples, the samples were immediately placed in a box on ice for cooling and were transported to the laboratory. After centrifugation at 500× *g* for 15 min at room temperature, the serum samples were frozen at −30 °C until further analysis.

### 2.2. SAA Concentration Measurement

SAA concentration was measured using an automated biochemical analyzer (Pentra C200, HORIBA ABX SAS, Montpellier, France; or Labospect 7180, HITACHI, Tokyo, Japan) with a special SAA reagent for animal serum or plasma (VET-SAA ‘Eiken’ reagent; Eiken Chemical Co. Ltd., Tokyo, Japan) based on a previous report with similar measurement conditions [12]. SAA concentration was calculated using a standard curve generated using a calibrator (VET-SAA calibrator set; Eiken Chemical Co. Ltd., Tokyo, Japan). Finally, the SAA concentration obtained from the two groups was used to determine the cut-off value of the SAA concentration in the JB breeding cattle group in our jurisdiction.

### 2.3. Data Management and Statistical Analysis

In the present study, a descriptive analysis was performed. Based on the statistical results, a logistic regression analysis was performed to obtain the cut-off values of the SAA concentration of each probability. Additionally, a proposed cut-off value of SAA concentration was determined to evaluate between healthy or diseased groups using a receiver operating characteristic (ROC) curve. The results of the SAA concentrations between clinically healthy and diagnosed as diseased cows were compared using the *t*-test. The results obtained for each herd are expressed as the mean ± standard deviation (SD). *p*-values less than 0.05 were considered to indicate statistical significance.

## 3. Results

The descriptive statistics revealed that the SAA in healthy cows had a lower mean concentration than the diseased cows (${\bar{x}}_{healthy}=2.77<54.8={\bar{x}}_{sick}$). Since the standard deviation (SD) for the healthy cow group was relatively small (*sd_healthy_* = 3.15), indicating a low variability, the SAA value for most healthy cows was close to the mean concentration. On the contrary, the SD for the diseased cow group was very large (*sd_sick_* = 76.5). As the difference between the mean value of healthy and diseased cows was only 52.03, the large value of the SD in the diseased cow demonstrates a large overlap of the SAA value between healthy and diseased cows. Since the data failed to satisfy the normality and homogeneity assumptions, the Mann-Whitney U (Wilcoxon rank-sum) test was used to compare the SAA between healthy and diseased cows. Nevertheless, we also considered the Student *t*-test due to the large data size. Table 2 shows a substantial difference between healthy and diseased cows.

Figure 1 shows the distribution pattern of the SAA concentrations derived from cows of both the healthy (*n* = 289) and diseased groups (*n* = 66). The mean SAA concentration in the cows of the healthy and diseased groups was 2.8 ± 3.2 mg/L and 54.8 ± 76.5 mg/L, respectively, and a significant difference was observed between the two groups (*p*-value = 6.68 × 10^−29^). The mean SAA concentration in the diseased group was approximately 20 times higher than that in the healthy group. The median SAA concentration in the cows of the healthy and diseased groups was 1.5 mg/L and 31.2 mg/L, respectively. The median SAA concentration in the diseased group was approximately 20 times higher than that in the healthy group.

Since a binary state was present, which is a disease state labeled “1” and a healthy state labeled “0”, logistic regression model analysis was performed. The disease/healthy state is a categorical response variable, and SAA is the predictor variable. The statistical analysis result revealed that the logistic regression model was significant (*p*-value < α = 0.05). Therefore, the obtained model could be used to predict disease in the cow based on the probability derived from the model. Having a certain value of SAA, we can get the probability of being disease for the cows in question.

Model:$$logit\left(p\right)=-3.74781+0.27297*SAA$$
$$p=\frac{e^{LO}}{1+e^{LO}}$$

Figure 2 shows the logistic regression curve and proposed diagnostic cut-off values of the SAA concentrations to differentiate between JB breeding cows of healthy and diseased groups based on the probability derived from the logistic regression analysis. The cut-off SAA concentrations at *p* = 0.05, 0.1, 0.5, and 0.9 were 2.9 mg/L (sensitivity: 0.9242424, specificity: 0.7785467), 5.7 mg/L (sensitivity: 0.8939394, specificity: 0.8719723), 13.7 mg/L (sensitivity: 0.6666667, specificity: 0.982699), and 21.8 mg/L (sensitivity: 0.6060606, specificity: 0.9965398), respectively. In the present study, as shown in Figure 3a,b, the proposed diagnostic cut-off point based on the ROC curve for SAA concentrations to identify healthy or diseased groups of JB breeding cows was set at 6.5 mg/L (*p* = 0.122).

Based on the SAA cut-off value obtained from the ROC curve (6.5 mg/L) in the present study, 31 out of 289 cows in the healthy group were estimated to possess an inflammatory status (≥6.5 mg/L) without clinical symptoms. On the other hand, in the diseased group, although eight out of 66 cows were estimated to have an inflammation and with disease by clinical veterinarians (emaciation: 1, hepatitis: 2, enteritis: 1, no appetite: 3, unknown: 1), however, the blood samples showed no inflammation (≤6.5 mg/L) of the blood samples. Nevertheless, recovery of the clinical symptoms was confirmed in all eight cows thereafter by the veterinarians.

## 4. Discussion

In the present study, we aimed to compare the SAA concentrations derived from both clinically normal and diseased cow groups measured by the LATIA method to confirm the practical usefulness of the method and determine the cut-off values of JB breeding cows in the serums of healthy and clinically diseased cows with some symptoms. Our results clearly indicated a significant difference (*p*-value = 6.68 × 10^−29^) between the healthy and diseased cow groups. To the best of our knowledge, this is the first verification test using the LATIA method for determining the cut-off values of the SAA concentration of JB breeding cows and the SAA concentrations in JB breeding cows.

Previous reports demonstrated that inadequate nutrition is one of the major factors responsible for prolonging calving-to-conception interval, which often causes subclinical and/or clinical metabolic diseases [15,16]. Therefore, we established a protocol of routine MPT for JB breeding herds. We could confirm that this improved the reproductive efficiency of the herd and established a monitoring database for perinatal health conditions (protein and energy metabolisms) and associated subclinical and reproductive disorders, in our region, Kagoshima Prefecture, Japan [17,18,19]. Furthermore, we recently reported that our established MPT-based evaluations were useful in the detection of sub-clinical abnormal metabolism in JB breeding herds resulting from not only low protein diets but also excessive high protein diets for improving the reproductive efficiencies [20]. Regarding herd health programs in cattle, the early detection of animals or herds at increased risk for sickness or compromised reproductive performance should be considered important factors for both farmers and veterinarians [4]. In order to achieve this goal, in addition to the nutritional monitoring of the herd using our established MPT, the identification of a biomarker associated with the non-nutritional stressors, especially an inflammatory marker, such as SAA, during the perinatal period of JB breeding cows may be useful in the prevention or early detection of perinatal diseases and improving reproductive efficiency, consistent with previous reports on dairy cattle herds [4]. The speed and practicability of an assay procedure are major criteria for accepting an analysis for routine estimation in clinical practice [9]. Recently, the SAA assay method with a LATIA reagent using an automated clinical chemistry analyzer was found to be superior in assessing the severity of inflammation in mastitis of dairy cattle and was considered a prompt method regardless of the number of the samples [12]. Therefore, in the present study, the incorporation of the SAA measurement in our routine MPT was possible by combining the LATIA method for SAA measurement with the biochemical test system using the auto-analyzer for simultaneous multi-sample measurement. SAA concentration has been considered both as a potential indicator of disease and well-being in individual animals and as an indicator of herd health [5,7,21,22]. Additionally, any effort to avoid the acute phase response in the transition period would be useful for optimizing the productive and reproductive performance in dairy cattle herds [13]. Thus, the establishment of MPT, including SAA measurement with the LATIA method in our routine MPT, could be useful in assessing both the nutritional and non-nutritional inflammatory status of JB breeding cows in the perinatal period, and possibly improve the reproductive efficiency of JB breeding cattle herds. Additionally, SAA concentration reportedly revealed significant differences among the examined farms [23]. Therefore, screening based on the cut-off SAA values obtained in the present study would help veterinarians identify diseased JB breeding cows or herds under regulatory inspection programs, such as MPT.

Based on the present study, if we selected an SAA cut-off value of probability 0.1, it would indicate that out of 100 cows with an SAA value of 5.7 mg/L, the model estimates approximately 10 % of the cows to have inflammation. Previous reports also indicated that although field veterinarians precisely identify healthy cows, which was also determined by the negative SAA results, they apparently lack the ability to identify cows with subclinical inflammatory responses whose serum reacts positively to SAA [24]. Indeed, in the present study, 31 cows diagnosed as healthy without any clinical symptoms indicated SAA concentrations more than the cut-off value (6.5 mg/L) obtained using the ROC curve, indicating that the 31 cows possibly had subclinical inflammation. Therefore, in addition to the results of SAA measurements, diagnosis of subclinical cases should be performed together with the results of other diagnostic markers, including other APPs. On the other hand, the SAA concentrations of eight cows out of the 66 diseased cows were ≤6.5 mg/L if we selected the cut-off value of 6.5 mg/L; however, the recovery of the clinical symptoms was confirmed in all eight cows thereafter by the veterinarians. These results clearly indicated that when introducing SAA concentration measurement in cattle in clinical practice, one measurement is not sufficient, and the continuous observation of the herd by a manager or veterinarian and, if available, evaluation of the degree of inflammation by re-measurement of the SAA in the cows are required.

## 5. Conclusions

In conclusion, our preliminary study indicated the cut-off values of SAA measured by the LATIA method in JB breeding cows, and suggested the usefulness of the measurement as a part of the MPT for JB herd management. However, not only further validation of the cut-off value of the SAA concentration with a large sample of cows but also comparisons of the sensitivity, specificity, and measurement cost with other APPs by the LATIA method are warranted in the future.

## Figures and Tables

**Figure 1 vetsci-09-00198-f001:**
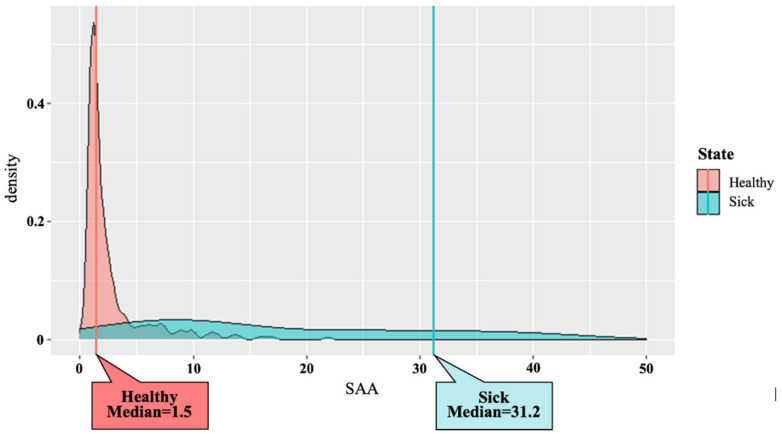
Distribution pattern of the serum amyloid A concentrations derived from cows in both healthy (*n* = 289) and diseased groups (*n* = 66).

**Figure 2 vetsci-09-00198-f002:**
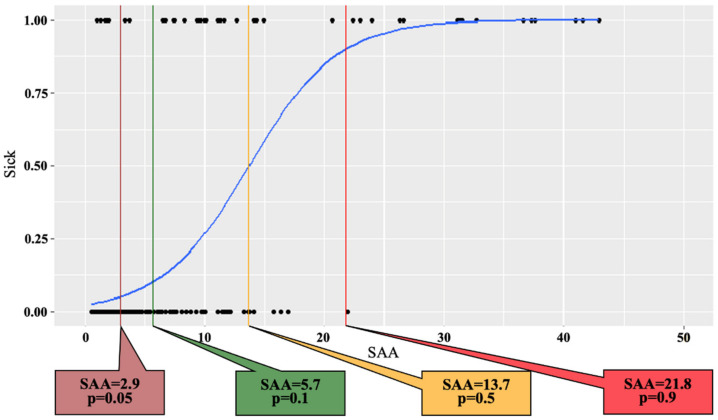
Proposed diagnostic cut-off points of serum amyloid A concentrations to differentiate between healthy (*n* = 289) and diseased (*n* = 66) groups of Japanese Black breeding cows evaluated with each probability (*p* = 0.05, 0.1, 0.5, and 0.9) by the logistic regression analysis.

**Figure 3 vetsci-09-00198-f003:**
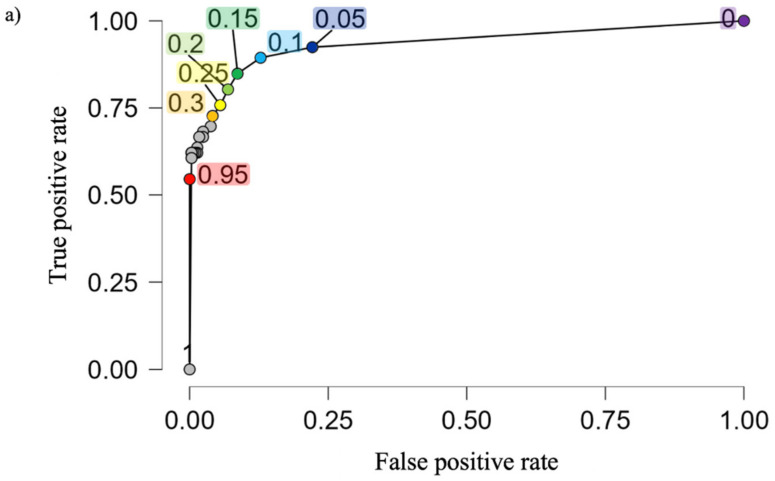
The receiver operating characteristic curves of the serum amyloid A (SAA) concentration to identify healthy or diseased groups of Japanese Black breeding cows. (**a**) representative cut-off values of each probabilities. Area under the curve: 0.940, *p*-value: 3.338084 × 10^−29^, Accuracy: 0.8901408, Sensitivity: 0.8787879, Specificity: 0.8927336, Precision: 0.6516854, Cut-off SAA value: 6.5 mg/L, Probability: 0.1220142, (**b**) Trade-off between Specificity and Sensitivity across the range of possible cut-offs.

**Table 1 vetsci-09-00198-t001:** Number of the diseased cows in the present study based on the clinical diagnoses by the veterinarians.

Clinical Diagnosis	No. of Cows
Reproductive disease	14
Fever, no appetite	11
Enteritis	10
Hepatitis	7
Lymphoma	7
Musculoskeletal disease	7
Unable to stand, downer	6
Urosis	3
Unknown	1
Total	66

**Table 2 vetsci-09-00198-t002:** The *t*-test comparison of the serum amyloid A concentrations between the healthy and diseased cow groups.

Method	y	Group 1	Group 2	Statistic	*p*-Value
Mann-whitney	SAA	Healthy	Disease	1153	6.68 × 10^−29^
Student	SAA	Healthy	Disease	−11.586	1.59 × 10^−26^

y: The variable being compared.

## Data Availability

The original contributions presented in the study are included in the article; further inquiries can be directed to the corresponding author.

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
