# Peer review of "Serum Amyloid A Concentrations of Healthy and Clinically Diseased Japanese Black Breeding Cattle—Preliminary Measurements for Determining the Cut-Off Concentrations"

_vetsci, 2022, doi:10.3390/vetsci9050198_

Round 1

Reviewer 1 Report

This paper introduces a study to find cut-off points assuming that the inflammatory state can be predicted by SAA concentration. This study is expected to define indicators for screening inflammatory status.  There is no description in the text on the sensitivity or specificity calculated for each cut-off point other than a figure of the ROC curve. There is a regret that the role of cut-off points cannot be detected. For examples, lines 28-30 are ambiguous.

In Discussion, the logical development of how these cut-off points can be used and what advantages the use has is insufficient. 

Lines 275-277 should be deleted, but it seems that autores forgot.

Author Response

Veterinary Sciences

Manuscript ID: vetsci-1643831

Title:

Serum amyloid A concentrations of healthy and clinically diseased Japanese Black breeding cattle – preliminary measurements for determining the cut-off concentrations

We have revised the manuscript in accordance with the suggestions of Reviewer 1 as follows. Both the revised sections of the manuscript and the responses to the Reviewer below are marked in red.

Specific Comments 

This paper introduces a study to find cut-off points assuming that the inflammatory state can be predicted by SAA concentration. This study is expected to define indicators for screening inflammatory status.  There is no description in the text on the sensitivity or specificity calculated for each cut-off point other than a figure of the ROC curve. There is a regret that the role of cut-off points cannot be detected. For examples, lines 28-30 are ambiguous.

In Discussion, the logical development of how these cut-off points can be used and what advantages the use has is insufficient. 

Thank you very much for your insightful comments. First, I would like to apologize for the lack of clarity. After the first submission, we have resubmitted a modified version with the results of ROC analysis added to obtain the optimum cut-off value in this study. Therefore, we believe that these concerns regarding the results and discussion, which you pointed out, were valid before this present round of modification. However, we have clarified these concerns in the revised manuscript. Although such a background is possible, accordingly, we have added the sensitivity and specificity for each cut-off point in the Results section. We would appreciate it if you could review the revised version of the manuscript. Lines 192-195.

Lines 275-277 should be deleted, but it seems that autores forgot.

Accordingly, we have deleted this part.

Reviewer 2 Report

Reviewer comments for manuscript ID vetsci- 1643831 entitled ‘Serum Amyloid a Concentrations of Healthy and Clinically Diseased Japanese Black Breeding Cattle – Preliminary Measurements for Determining the Cut-off Concentrations’

General Comments

It is a useful study on determination of biomarker concentrations in healthy and diseased cattle. The work is contemporary, focussed on the prevention and early diagnosis of pathological processes in cattle for improved breeding efficiency and production using a biomarker. Introduction clearly identifies the gaps in literature and provides a fair idea about the research. Data is sufficient to draw preliminary conclusions. Statistical methods used are appropriate and robust. Graphical representation of results adds to the quality of the manuscript. Discussion is very superficial. I would like to see the pathological basis of SAA being used as a biomarker for the benefit of the readers. As claimed by the authors, please explain how SAA is more sensitive, specific, and reliable than other biomarkers. For the assertion, costs notwithstanding, other biomarkers should have been analysed by this LATIA method for more comprehensive study and scientifically valid conclusions. It is fine to work on one biomarker, but then the conclusions/assertions should be more guarded. I would like to see the corrections done based on the suggestions before I recommend the publication of the paper.

Specific Comments

Line 2: Please write ‘A’ instead of ‘a’.

Line 23: Please write ‘serum samples’ instead of ‘serums’

Line 46: Please delete ‘identifying’

Line 55: Please replace ‘estimating’ with ‘identifying’

Line 65: Please replace ‘fields’ with ‘practice’

Line 65: Please delete ‘effects’

Lines 66-67: Please reword ‘the SAA measurements for only experimental purposes in cattle have been performed so far’ as ‘the SAA analysis in cattle has been restricted to experimental purposes only’

Lines 67-68: Please delete ‘Moreover, they have not been applied in actual bovine routine clinical practice, such as within the metabolic profile test (MPT) yet’

Line 79: Please replace ‘pathological’ with ‘diseased’

Line 84: Please reword ‘…disease prevention of the cattle herd’ as ‘…disease prevention and early treatment of the cattle herd’

Line 92: Please delete ‘in our jurisdiction’

Lines 224-25: Please delete ‘indicating at each probability in our jurisdiction’

Lines 228-34: Please reword these sentences as ‘Therefore, we established a protocol of routine MTP for JB breeding herds. We could establish that this improved the reproductive efficiency of the herd and established a monitoring database for perinatal health conditions (protein and energy metabolisms) and associated subclinical and reproductive disorders, in our region of Kagoshima Prefecture, Japan (Watanabe et al., 2013a, b; Watanabe et al., 2014).

Lines 244-45: Please reword ‘…. may be useful in preventing perinatal diseases and improving reproductive efficacy’ as ‘may be useful in prevention or early detection of perinatal diseases and improving reproductive efficacy’

Author Response

Veterinary Sciences

Manuscript ID: vetsci-1643831

Title:

Serum amyloid A concentrations of healthy and clinically diseased Japanese Black breeding cattle – preliminary measurements for determining the cut-off concentrations

We have revised the manuscript in accordance with the suggestions of Reviewer 2 as follows. The responses to the Reviewer below are marked in blue.

General Comments

It is a useful study on determination of biomarker concentrations in healthy and diseased cattle. The work is contemporary, focussed on the prevention and early diagnosis of pathological processes in cattle for improved breeding efficiency and production using a biomarker. Introduction clearly identifies the gaps in literature and provides a fair idea about the research. Data is sufficient to draw preliminary conclusions. Statistical methods used are appropriate and robust. Graphical representation of results adds to the quality of the manuscript. Discussion is very superficial. I would like to see the pathological basis of SAA being used as a biomarker for the benefit of the readers. As claimed by the authors, please explain how SAA is more sensitive, specific, and reliable than other biomarkers. For the assertion, costs notwithstanding, other biomarkers should have been analysed by this LATIA method for more comprehensive study and scientifically valid conclusions. It is fine to work on one biomarker, but then the conclusions/assertions should be more guarded. I would like to see the corrections done based on the suggestions before I recommend the publication of the paper.

Thank you very much for your insightful comments. The main purpose of our preliminary study was to obtain the cut-off values of the SAA concentration in JB breeding cattle by the LATIA method, and we agree with your comment that it is necessary to compare the sensitivity, specificity, and measurement cost with other APPs measurement systems by the LATIA method. Therefore, following your comment, we have added in the conclusion parts that it is necessary to evaluate the usefulness of SAA in bovine in clinical practice by comparing it with other APPs in the future. Lines 287-290.

Specific Comments

Thank you very much for your comments. We have revised the manuscript according to your comments.

Line 2: Please write ‘A’ instead of ‘a’.

We have corrected this accordingly. Line 2.

Line 23: Please write ‘serum samples’ instead of ‘serums’

We have revised this accordingly Line 23.

Line 46: Please delete ‘identifying’

We have deleted the word “identifying”.

Line 55: Please replace ‘estimating’ with ‘identifying’

We have replaced the word “estimating” with “identifying” Line 55.

Line 65: Please replace ‘fields’ with ‘practice’

We have replaced the word “fields” with “practice” Line 65.

Line 65: Please delete ‘effects’

We have deleted the word “effects”.

Lines 66-67: Please reword ‘the SAA measurements for only experimental purposes in cattle have been performed so far’ as ‘the SAA analysis in cattle has been restricted to experimental purposes only’

We have revised the sentence accordingly. Line 66.

Lines 67-68: Please delete ‘Moreover, they have not been applied in actual bovine routine clinical practice, such as within the metabolic profile test (MPT) yet’

We have deleted this sentence.

Line 79: Please replace ‘pathological’ with ‘diseased’

We have replaced the word “pathological” with “diseased” Line 78.

Line 84: Please reword ‘…disease prevention of the cattle herd’ as ‘…disease prevention and early treatment of the cattle herd’

We have replaced the sentence accordingly. Lines 82-83.

Line 92: Please delete ‘in our jurisdiction’

We have deleted this phrase.

Lines 224-25: Please delete ‘indicating at each probability in our jurisdiction’

We have deleted this phrase.

Lines 228-34: Please reword these sentences as ‘Therefore, we established a protocol of routine MTP for JB breeding herds. We could establish that this improved the reproductive efficiency of the herd and established a monitoring database for perinatal health conditions (protein and energy metabolisms) and associated subclinical and reproductive disorders, in our region of Kagoshima Prefecture, Japan (Watanabe et al., 2013a, b; Watanabe et al., 2014).

We have revised the sentence accordingly with small modification. Lines 231-234.

Lines 244-45: Please reword ‘…. may be useful in preventing perinatal diseases and improving reproductive efficacy’ as ‘may be useful in prevention or early detection of perinatal diseases and improving reproductive efficacy’

We have revised the sentence accordingly with small modification. Lines 245-246.

Round 2

Reviewer 1 Report

In the text, probability and P-values must be clearly distinguished.   Expression of 'p=' in the abstract (line 30) is not well distinguished whether it is p-value or calculated probability. The four vertical lines do not show a clear description. I think it would be good to indicate each point  that the horizontal axis SAA value and vertical axis probability meet.

Author Response

Veterinary Sciences

Manuscript ID: vetsci-1643831

Title:

Serum amyloid A concentrations of healthy and clinically diseased Japanese Black breeding cattle – preliminary measurements for determining the cut-off concentrations

We have revised the manuscript in accordance with the suggestions of Reviewer 1 as follows. Both the revised sections of the manuscript and the responses to the Reviewer below are marked in red. Additionally, the responses to the Academic Editor Comments. 

Comments 

In the text, probability and P-values must be distinguished. Expression of 'p=' in the abstract (line 30) is not well distinguished whether it is p-value or calculated probability. The four vertical lines do not show a clear description. It would be good to indicate that the horizontal axis SAA value and vertical axis probability meet at each point.

Thank you very much for your insightful remarks. According to your comments, we have distinguished between “probability” and “P-value” in the text. Lines 27, 159, 225.

Regarding four vertical lines, our original WORD file clearly shows it, but in the file converted to a PDF file, the cause is unknown, but the arrow disappeared. I would like to make corrections at the editing stage.  

Response to the Academic Editor's Comments

It is significant to reveal early diagnosis markers in the clinical background to help with early diagnosis and treatment. This study was aimed to use SAA as a test marker to differentiate diseased from clinically healthy cattle and determined that SAA could be used as a diagnostic marker. For the clinical point, the most important issue is the early detection of diseased animals before they become clinical diseases. However, the experiment lacked the essential control of subclinical animals. Therefore, this group of subclinical animals would challenge the sensitivity and specificity obtained in this study. Although the authors stated this work was preliminary measurements, I still hope the authors will get more data about subclinical animals and then submit their papers.

Thank you very much for your valuable and insightful comments. Accordingly, we will get more data focusing on the sub-clinical animals in the practical fields and would like to submit our new manuscript. Thank you very much again. 
